# Job Seekers’ Burnout and Engagement: A Qualitative Study of Long-Term Unemployment in Italy

**DOI:** 10.3390/ijerph20115968

**Published:** 2023-05-26

**Authors:** Marcello Nonnis, Mirian Agus, Gianmarco Frau, Antonio Urban, Claudio Giovanni Cortese

**Affiliations:** 1Department of Pedagogy, Psychology, Philosophy, University of Cagliari, 09123 Cagliari, Italy; mirian.agus@unica.it (M.A.); gmf871@gmail.com (G.F.); 2Cagliari University Hospital, 09124 Cagliari, Italy; a.urban@aoucagliari.it; 3Department of Psychology, University of Turin, 10124 Turin, Italy; claudio.cortese@unito.it

**Keywords:** long-term unemployment, job seeking, burnout, engagement, disillusion, qualitative study, JD-R model of burnout

## Abstract

Long-term unemployment has major consequences from an economic, physical and psychosocial perspective. Several authors have pointed out that the search for employment is in itself work, which can generate feelings of exhaustion of psychophysical energies, cynicism and disinvestment, as well as a sense of ineffectiveness to the point of complete disillusion. The construct of burnout can be used to describe this psychological process. This study evaluated the burnout and engagement dimensions in individuals searching for work for a long time, from a qualitative perspective. Fifty-six semi-structured interviews were conducted with a sample of long-term unemployed job seekers (Sardinia, Italy), based on Maslach’s model of burnout (exhaustion, cynicism, effectiveness in job search). The answers to the semi-structured interviews were processed through T-Lab, a semi-automatic textual analysis software. Four thematic cores emerged: exhaustion vs. engagement, cynicism vs. trust, inefficacy vs. efficacy in job search and disillusion vs. hope. This result is consistent with the four-dimensional theoretical model of burnout, originally proposed by Edelwich and Brodsky, recently taken up by Santinello, and framed as the opposite of engagement, as shown in the JD-R model. This study highlights that burnout can describe the psychosocial experiences of long-term unemployed job seekers.

## 1. Introduction

### 1.1. Unemployment in the Psychological Perspective

Unemployment is a major problem and is systematically a feature of governments’ social and economic policies. In 2019, the unemployment rate in Europe stood at 6.3%; however, there is a wide diversity among EU countries, and in the same year Italy recorded a figure of 9.9% [1]. Since the early 2020s, the COVID-19 pandemic has led to an overall rise in unemployment [2], and its effects on employment are still unfolding [3].

The Italian government has taken several measures, some of which have been agreed upon at the European level [4], such as stopping layoffs and supporting the many workers and companies affected by the pandemic [5]. However, it is estimated that the impact of the pandemic on unemployment will increase and will also affect a part of the inactive population, due to the restrictions still in place and the current climate of job insecurity [6,7].

Unemployment has important consequences in several respects. According to the literature, psychological consequences can include: suicide risk; psychiatric disorders; impaired mood stability (depression, anxiety, demoralization, frustration, experiences of anger, guilt, inadequacy, hopelessness and pessimism [8,9]); cognitive problems (difficulty in concentration, lower mental efficiency [10]); and feelings of loneliness or social isolation [11]. Serious problems with self-esteem and impairment of personal aspirations have also been noted [12].

Physical symptoms related to unemployment [13] include cardiovascular, immunological, dietary and gastrointestinal symptoms, respiratory and biochemical disorders, asthenia, headaches, and sleep and sexual problems. A state of prolonged unemployment also has important consequences from a social point of view. Indeed, there is an increase in separations, divorces, crime, alcohol or drug abuse and a decline in the birth rate among the unemployed, together with a sense of loss of status and social utility [14].

Some authors have suggested that the relationship between long-term unemployment and the strategies for finding a new job are more refined, complex and multifaceted than the mere relationship between unemployment and its consequences that have thus far been discussed. McKee-Ryan et al. [13] found that during unemployment, the centrality of work for individuals is negatively correlated with their mental health and life satisfaction, while the social threat associated with the status of being unemployed and the financial strain due to the erosion of economic resources negatively affect the well-being of unemployed people. The availability of financial resources and diverse coping strategies (such as the use of social support or the ability to structure time) are positively correlated with the well-being of the unemployed. Negative evaluations of unemployment also negatively affect their well-being, while expectations of future re-employment have a positive influence on it. However, according to McKee-Ryan et al. [13], active engagement in searching for employment correlates negatively with people’s mental health, as searching for a job is a stressful activity. They found that unemployment benefits give less protection than would be expected against the negative effects of unemployment. In fact, the decline in the well-being of the unemployed is not solely due to financial deprivation, because the quality of work and the value people place on work are also very important factors. 

Further confirmation of this is provided by the study by Kossen and McIlveen [15], who see unemployment as the state of being deprived of decent work. The main predictor they considered was marginalization, which leads to decreased aspirations and worse access to decent work. Marginalization also places structural limits on the social mobility of the long-term unemployed, with several negative psychological effects including reduced educational opportunities for their families. From this perspective, high self-efficacy [16] among the unemployed can, paradoxically, become a hindrance: if the socioeconomic environment does not respond to the aspirations of the unemployed, there is a risk that they will experience an intense sense of failure or frustration over time.

The paradoxical effect that agency can have on the psychosocial malaise of the unemployed seeking re-employment has also been highlighted by Duffy et al. [17], according to whom having a high level of proactivity increases one’s chances of finding a job, but if one is unable to access decent work within a short time, there is a negative impact on volition itself.

Some researchers have analysed the role played by the main dimensions of psychological capital on unemployment [18]. Merino et al. [19] highlighted the influence of environmental mastery, vitality and resilience on the eustress of the unemployed, as well as the moderating role of environmental mastery and optimism on their distress. According to Fernández-Valera et al. [20], if an unemployed person has an adequate level of self-efficacy, this can mitigate the negative psychological processes that may affect his or her psychophysical health.

Lim et al. [21] analysed the relationship in unemployed job seekers between fatigue and tension over their financial status, their social marginalization and their psychological capital. Their results showed that financial tension is positively correlated with fatigue, as people have to divide their resources with respect to two different coping strategies: the first aimed at finding a new occupation and the second focused on managing contingent difficulties. Social marginalization is positively correlated with fatigue, as sociality has an instrumental role for the unemployed in their search for an occupation, as well as being an emotional support and exercising a time-regulation function. Psychological capital has also a negative relationship with fatigue, as it is a personal resource that provides the energy needed to cope with search activities [21]. 

In our view, these studies clearly demonstrate the opportunity to frame the dynamics of the relationships between long-term unemployment, the search for new employment and the consequences of both these situations, within a more articulated psychosocial construct.

### 1.2. Job Burnout

Burnout is an occupational psychosocial syndrome that is characterized by three dimensions: (a) feelings of emotional or psycho-physical exhaustion; (b) increased mental distance from one’s work, with feelings of negativity or cynicism; and (c) reduced professional effectiveness. Originally, Maslach et al. [22] defined burnout as a degenerative work-related stress syndrome resulting from the emotional strain caused by dealing with people in need of help. Recently, job burnout was included among nonmedical conditions in the 11th revision of the International Classification of Diseases (ICD-11) [23,24].

Over the past two decades, burnout has been extensively framed as a work-related organizational pathology in service settings, within the job demands–resources (JD-R) model of job stress [25,26,27,28], more appropriately taking the name job burnout. According to this model, job burnout and work engagement [29] are considered opposite patterns, and the latter also consists of three dimensions: vigour is connoted by high mental energy and resilience while working and persistence; dedication is a sense of meaning, enthusiasm and pride toward one’s work; and absorption is characterized by being completely focused on one’s work [30,31,32].

According to the JD-R model, job demands (such as time pressure, inadequate physical environment or excessive workload) and job resources (such as being able to exercise adequate control over one’s work, being able to participate in decision making, or perceived organizational support) are present in organizational contexts, and job burnout results from a combination of excessive work demands and inadequate job resources [27,30]. This dynamic results in the depletion of mental and physical resources, job disengagement and a combination of mental, physical, motivational and relational consequences.

The results most frequently found in the literature can be traced back to the three stages previously described. Exhaustion involves a collapse of psychophysical energies, with symptoms typical of anxious–depressive states (resistance to engaging in an activity, apathy, demoralization, difficulty in concentrating, discomfort, sleep disturbances, mood alteration, intense and unmotivated worrying, and feelings of inadequacy, guilt, frustration and failure) [22,33]. In the cynicism dimension, there is a strong decrease in motivation and a decline in work engagement, emotional detachment from work, pessimism and hostility towards colleagues and users [34,35]. As regards professional ineffectiveness, the worker manifests a decline in self-esteem, a distrust in their own abilities and resources, feelings of inadequacy towards work and a decreased desire for success [36,37].

More recently, some authors [38,39] have reiterated the importance of the self-actualizing, vocational and value aspects of employment with respect to the syndrome, and have identified disillusion in addition to the three dimensions mentioned above. This last stage of job burnout was already present in the defining models of Edelwich and Brodsky [35,40,41] and requires one to consider the meaning that work has for the person in society and for his or her own existence. Therefore, disillusion in burnout represents the attrition and impairment of professional ideals and job expectations. 

Finally, research on job burnout has further broadened its scope, and the syndrome has proved to be effective in describing distress in additional areas of human activity, such as study [42], competitive sports [43] and volunteering [44].

### 1.3. The Burnout of Unemployed Job Seekers

According to several scholars, the job search is work in itself and can lead to feelings of exhaustion or depletion of psycho-physical energy, to experiences of cynicism and disinvestment from the search; it can also lead the unemployed person to experience a sense of ineffectiveness, to the point of complete disillusionment—all the more so if the unemployment lasts for a long period. This process is similar to burnout [45,46,47].

For example, Amundson and Borgen [45,48] described the emotional roller coaster that the unemployed may experience in their search for new employment. In their studies, burnout is described by the model of Edelwich et al. [38,41]: in fact, after the grieving phase due to job loss, people invest their energy, enthusiasm and (sometimes unrealistic) expectations in the search for a new occupation. If, however, this search proves fruitless, the unemployed person experiences stagnation, frustration and anger, which sometimes result in maladaptive behaviour (such as alcohol or drug abuse). If the search continues unsuccessfully, the unemployed person falls into a state of apathy, during which he or she devotes a decreasing amount of energy to the search, falling into a downward spiral of disillusion [45,48].

Nonnis et al. [49], in a recent quantitative exploratory study conducted using Leiter and Maslach’s Organizational Check-up System (OCS) burnout questionnaire [50] adapted to long-term unemployed job seekers (Italian sample, *n* = 208), showed that this construct can describe the psychosocial experience of those in this condition. Their study hypothesized that the construct of burnout consists of the three dimensions of the model developed by Maslach et al. (exhaustion, cynicism, ineffectiveness in job search [22,25]); but the results suggested that the “jobless burnout” construct can best be articulated into four dimensions: psychophysical exhaustion and the ineffectiveness in job seeking are confirmed, but the dimension of cynicism is divided into two distinct factors. The first factor can be called disengagement from the possibility of finding an occupation, whatever it may be. The second factor denotes disillusion with the possibility of finding a job consistent with one’s motivations, interests, aspirations and qualifications. This result is in line with the descriptive models of the burnout syndrome proposed by Edelwich et al. and taken up more recently by other authors [35,38,39,41]. In addition, the length of the search period correlates positively with burnout and disillusionment disengagement and negatively with search effectiveness. Moreover, older unemployed people consider themselves to be less effective and more exhausted from the job search than younger unemployed people.

This result highlights that burnout leads to psycho-physical exhaustion, detachment from one’s work and to the wearing down (to the point of impairment) of the professional ideals of the unemployed person to the point of complete disillusionment. Based on what has been argued so far, it is plausible to use the construct of job burnout as a framework for understanding the psychosocial experience of unemployed seekers. However, in the literature, to the best of our knowledge, only a few works have assessed the dimensions of burnout and engagement in long-term job seekers using a thematic analysis methodology.

This study therefore proposed to evaluate the meaning of burnout and engagement in long-term unemployed job seekers from a qualitative perspective, with reference to the three-dimensional model [22,23,24,25] (exhaustion, cynicism, effectiveness in job search) adapted to unemployed job seekers.

## 2. Materials and Methods

### 2.1. Research Design

The present study is qualitative and semantic in nature. It was conducted with part of the sample of unemployed job seekers who were included in the previous study [49] described above. The study was conducted through semi-structured, face-to-face, audio-recorded interviews in Italian that were then transcribed in full using word processing software. The group of participants was identified based on the availability of interviewees, using non-probability sampling for representative items (purposive sampling [51]) and following a multiple case study approach. The data were collected in the first two months of 2020 (just before the first lockdown in Italy due to the COVID-19 pandemic).

### 2.2. Instruments

The research protocol is divided into two parts. The first includes a section collecting sociodemographic data: gender, age, education level and duration of seeking new employment. The second part involves a semi-structured interview consisting of six questions, two for each of the three dimensions of burnout and engagement investigated, applying the three dimensional model [22,25] adapted to long-term unemployment: psycho-physical exhaustion vs. engagement in job search; cynicism vs. engagement in job search; and ineffectiveness vs. effectiveness in job search. In each pair of questions, as reported below, the first explores the negative and discomfiting aspects of job search (burnout), while the second examines the positive and motivating ones (engagement). 

The semi-structured interview questions, asked of the respondents, are as follows:

Engagement/Exhaustion

-Engagement) What makes you feel energetic when looking for work?-(Exhaustion) What makes you feel exhausted when looking for work?

Involvement/Cynicism

-(Involvement) What makes you feel involved in your job search?-(Cynicism) What makes you feel detached in your job search?

Effectiveness/ineffectiveness.

-(Search effectiveness) What makes you feel effective in your job search?-(Search ineffectiveness) What makes you feel ineffective in your job search?

### 2.3. Participants

The sample was recruited, on a voluntary basis, from among the users of three employment centres in Sardinia (Italy). Starting from the first study by Nonnis et al. [49], qualitative data from the textual responses of 56 participants who agreed to be interviewed were analysed. Among the sample, 64.3% (*n* = 36) were female (males *n* = 20; 35.7%); participants had an average age of 37.41 years (SD = 11.48); they had, on average, been searching for a job for 46.27 months (SD = 54.60). Regarding the participants’ level of education, 14 participants (25.0%) had attended primary and secondary school, 30 (53.6%) had obtained a high school diploma, 2 (3.6%) had a bachelor’s degree and 10 (17.9%) had a master’s degree or higher educational qualification. All participants gave their response to all questions (there were no missing data).

### 2.4. Ethical Issues

The study was conducted in accordance with the guidelines of the Declaration of Helsinki and was authorized by the Ethics Committee of the University of Cagliari (Approval Number 0179701, 25 August 2021) in full compliance with the Ethical Principles of Psychologists and the Code of Ethics of the American Psychological Association (APA) and of the Italian Psychological Association (AIP). The study did not address sensitive topics and was conducted through procedures for informed and consenting adults. Finally, in accordance with Italian privacy law, the research guaranteed the anonymity and confidentiality of all participants.

### 2.5. Statistical Data Analysis

A qualitative approach was applied [52,53,54,55]. The qualitative analysis of the text produced through the semi-structured interviews was based on the case study. The text produced in the answers was evaluated by a software for lexicometric analysis, which assesses the frequencies with which words are used within sentences. This lexicometric analysis was conducted with reference to a specific variable of interest (such as the areas investigated in the questions).

In the literature, different authors [56,57] have identified essential aspects of the lexicometric approach: frequency analysis for every word (occurrence) of the vocabulary of the collection was applied to identify important terms; concordance analysis was conducted to examine local contexts/sentences of interest (these results were analysed as “key-word in context”); identification of characteristics of sub corpora, which were chosen by meaningful criteria (e.g., different categories of individuals, specific questions, defined themes); and co-occurrence analysis to examine significant contexts of terms on a global textual corpus level. 

These approached made it possible to apply a statistical test establishing which words occurred together more frequently within the corpus than purely by random likelihood. A further step of lexicometric analysis refers to the application of multivariate methods (e.g., to identify clusters of co-occurring words, evaluating the importance of specific words for a given question/theme/category of participants).

The answers to the semi-structured interviews were transcribed verbatim. The text was analysed using T-Lab software (release 8.0) [58], which allowed the application of semiautomatic textual analysis [55,59]. To perform the textual analysis, the answers were organized in relation to the above-mentioned dimensions [22,50]. The T-Lab software applied semiautomatic lemmatization to the recognized words. The researcher supervised these procedures, revising words unknown to the software (i.e., words that were not included in the T-Lab dictionary). 

This revision allowed, for example, for the clarification of the meaning of homograph words (which hypothetically might have several connotations). At this point, the lemmatized text was evaluated by the software to consider the frequencies of the use of words in the full corpus and in distinct sub-sections of the text (identified on the basis of the theoretical dimensions under investigation).

After the preparation of the corpus, cluster analysis was carried out to identify the contextual fields of meaning that recurred most often in respondents’ answers; this procedure allowed the identification of the recurring themes that were common among the responses [59]. The software shaped each cluster, putting together words that were often used in the same sentences (elementary contexts); these words were classified by reference to their chi-squared value [58]. Additionally, the T-Lab software detected several relevant descriptive variables, which were useful for defining each cluster; the clusters, the words and the descriptive variables were inserted in a factorial plane, identified by the application of binary lexical correspondences analysis [60,61].

## 3. Results

The corpus counted 15,161 words (tokens) of 2475 types. The process of customization of the vocabulary was conducted on 2049 lemmas; the number of hapax (lemmas used in the corpus only once) was 1361. A threshold of four occurrences was defined to find the key words to run the data analysis (*n* = 204 key words). Cluster analysis was applied using the bisecting K-means algorithm [58,62]. The analysis emphasized four clusters (Figure 1, Table 1), that considered the subsets of the corpus defined as “context units” (CUs); in each CU, sentences or paragraphs were identified and related to the bipolar dimensions (positive and negative) of the semi-structured interviews.

Cluster 1, Exhaustion vs. Engagement (21.70% of CUs), includes the highest number of CUs; it refers to relevant aspects for the work of practitioners, comprising positive and negative aspects. We can identify words such as “work”, “pleasure”, “thinking” and “profession”. These themes are related to the dimensions of cynicism and effectiveness in job search (see Table 1 and Figure 1). Regarding the negative polarity of this cluster, some typical sentences are:

“[…] not being able so far to do it professionally is what brings me back instead”.

“[…] looking here rather than running straight out, which is an option anyway […]”.

“[…] my profession—I have never done it, professionally, but only as a volunteer. It’s not like I have to be forced to do a job just to make a living”.

In contrast, some typical sentences with a positive polarity of Cluster 1 are: 

“[…] I’m still looking for work that will allow me to then get to my profession”.

“[…] I’m still passionate about what I do […] I really like what I’ve studied, so I would like […] to try to put it into practice, and so I seriously believe that I can change things in my own small way”.

“[…] the idea of being able to find a job in the field that I like is what makes me live peacefully. It is the spring that drives me to keep doing it”.

“[…] I was lucky enough to do the job I liked for 30 years. I chose it […] I was one of the few who got up in the morning happy to go to work […]”.

In this semantic core, the polarities of Exhaustion vs. Engagement appear as two sides of the same coin, as they testify to the hard work and motivational capacity of trying to make one’s work and profession an interest or an educational path that one has accomplished or undertaken with determination and passion. Thus, the respondents are searching for a job not only to sustain but also to actualize themselves.

Cluster 2 might be called Disillusion vs. Hope (10.83% of CUs); it includes words such as “answering”, “curriculum”, “sending”, “announcements”, “understanding”, “absence” and “usual”. These lemmas are related to the dimension of Exhaustion (see Table 1 and Figure 1). Some representative sentences of the negative polarity of this cluster are: 

“[…] the constant refusals! […] classic doors in the face!”

“[…] always do the same thing! That is […] turn on the computer, send the CV, wait. Send your resume and wait. At least, both, I don’t know, I’ll be the one who will be unlucky […] in the end it’s this: having to wait”.

“[…] When people immediately close the door on you, without giving you a thread of hope”.

“[…] Looking for work and having your hands cut off.”

“[…] Very worried, because […] my son […] is almost 12 years old […]. I don’t have the money to buy him a sandwich”.

A representative sentence of the positive polarity of this cluster is:

“[…] So maybe here […] maybe at the beginning you’re all charged up: come on, I have this, I managed […] Do, send [your CV], send here, send there! Come on, go, they’ll call you […]”.

In this second thematic core, the Disillusion polarity focuses on the malaise caused by job search strategies that repeatedly prove to be ineffective, the difficulty of creating diversified and innovative problem-solving strategies (as opposed to strategies that have proved to be ineffective) and the structuring of an external locus of control [63] with respect to the goal of finding work (e.g., passively waiting for something to happen or perceiving oneself as unlucky). Moreover, the theme of passive waiting—in this case hopeful waiting—is also present in the positive polarity (Hope) of the thematic core, while, interestingly, very current and significant themes for job success, such as that of job crafting [64] (the ability to create, invent and experiment with a job activity), are completely absent.

The third cluster could be defined as “Cynicism vs. Trust” (12.04% of CUs), and has the following lemmas: “finding”, “hope”, “possibility”, “vigorous” and “positive”. These words imply the dimension of Exhaustion (see Table 1 and Figure 1). The following sentences are representative of the negative polarity of this cluster:

“[…] I haven’t found anything so far. And above all, I failed to assert my credentials. What I have or maybe what I’m worth hasn’t helped me at all. I did a year off”.

“[…] I sometimes have difficulty in being able to sign up in some sections of the (public, n.d.a.) competition […], I can’t enter the posting”.

For the positive polarity of this cluster, we can count the following typical sentences:

“[…] the feedback! […] I find something that interests me and I can apply! Definitely, it’s positive!”

“[…] When I’m motivated to look for work in a sector that I like anyway, and I know it wouldn’t feel heavy”.

In the third thematic core, the polarities of Cynicism vs. Trust also appear to be complementary, and they are associated with the theme of the feedback—negative or positive—that is received by unemployed job seekers from the world of work regarding what they feel they can do and also what they feel they are worth, through the work they would like to do, if given the opportunity.

The fourth cluster (55.43% of CUs) depicts aspects related to “Inefficacy vs. Efficacy”, including lemmas such as “searching”, “experience”, “succeeding”, “working”, “seek”, “putting”, “asking”, “arriving”, “taking”, “ability”, “difficulty” and “situation”. These aspects seem to characterize the text regarding the negative and positive aspects of Effectiveness in Job Search and Cynicism (see Table 1 and Figure 1). Some representative sentences of the negative polarity of this cluster are: 

“[…] Our problem is that the possibilities are few”.

“[…] you are rejected because there is another who has one comma more on the curriculum than you!”

“[…] Well, it’s just a matter of numbers! Not of ability but purely of probability! Because there are so many unemployed! […] Companies then make a weapon of this! They try to be as selective as possible”. 

“[…] Maybe you should also take courses, to update yourself, hence the fear of needing them to be able to integrate”.

“[…] there is an unhealthy competitive climate, […] if they take you, it’s only to use you with internships or […] you’re low paid and actually they don’t even train you!”

Regarding the positive polarity of this cluster, we can count the following typical sentences:

“[…] Capable and knowing me, definitely and also […] wishing and having an expectation, which then, in the end […] will be fine!”

“[…] Confidence in my abilities then! I’m confident it’s more likely to go well, right, and then, try again! I want to try and go as it goes!”

With regard to both the negative and the positive polarities of the fourth core (Inefficacy vs. Efficacy in job search), experiences of the passive expectation of being chosen or selected from among many other competitors also emerge. It is only about the negative polarity that the fear emerges of a position of great contractual weakness, susceptibility to blackmail and exploitation by possible employers. Also sharply interesting is the (partly illusory) expectation that, per se, training or further education will provide better chances of job placement. There also emerges an equally passive (and perhaps illusory) expectation in the positive polarity that things will work out, in the form of wishing, hoping and betting on oneself, but without reference to more articulate and reasoned coping strategies for dealing with the job search.

## 4. Discussion

The effects of unemployment are known and documented in the scientific literature, but the psychosocial experiences of those who, in a condition of medium- or long-term unemployment do not give up and continue their efforts and engagement to search for a job, possibly a decent one, have not been extensively studied. As we have argued, studies centred on welfare loss [12,13,15,46] or on the constitutive dimensions of psychological capital [19,20] have highlighted some aspects of this issue, but this study, along with the previous one by Nonnis et al. [49], might, through the construct of “jobless burnout”, propose an articulated, comprehensive and processual key to interpreting the psychosocial experiences of those who are unemployed and engaged in a job search.

In general, the results suggest that although the semi-structured interviews were conducted based on Maslach and Leiter’s [22,50] theoretical model of burnout (psychophysical exhaustion, cynicism, and job search ineffectiveness) and the related positive engagement counter-polarities of the model of Schaufeli et al. [25,28], four thematic cores that emerged from the elaborations of the text corpus: Exhaustion (vs. Engagement), Disillusion (vs. Hope), Cynicism (vs. Trust) and Inefficacy (vs. Efficacy) in the job search. This reaggregation of the experiences of long-term job seekers is consistent with the theoretical model of burnout initially proposed by Edelwich et al. [41] and recently taken up by Santinello et al. [38], who integrated and updated these four negative dimensions of burnout with corresponding positive counter-polarities related to the engagement construct [31].

This result is also consistent with studies of unemployed job seekers by Amundson and Borgen [45,48], and the recent quantitative study by Nonnis et al. [49], who equally converged on the Edelwich et al. [41] model of burnout. A plausible explanation for the ability of the four-dimensional model of Edelwich et al. to better describe the burnout of job seekers than the three-dimensional model [22,25] may be found in the fact that, in the latter, the first considered dimension, that of exhaustion, in some ways does not take into account the preceding phase of commitment and (sometimes illusory) hope that animate those who have lost their jobs and are active in the search for new employment. Instead, this stage is considered by Edelwich et al. [41], who in fact called it “idealistic enthusiasm”, and this is reflected in the experiences of the interviewees.

The model developed by Maslach et al. [22,25] also considers the last stage to be that of professional ineffectiveness, while Edelwich and Brodsky’s model [41] goes further, also giving importance to the self-actualizing, value and motivational aspects of work for the person, as well as the ability of burnout to compromise them. In this sense, the search not for just any job, but for an occupation that is able to satisfy and fulfil them, seems to have relevance for unemployed job seekers, and burnout can undermine these (otherwise legitimate) aspirations.

On the other hand, looking at the positive counter-polarities of “jobless burnout”, attributable to the dimension of work (search) engagement that emerged in the thematic cores of the interviews, these are topical and in line with the studies of Schaufeli et al. [28,29,30], who have long pointed out that burnout and work engagement are two sides of the same coin. The results of our study suggest that the relationship between these two constructs can also be considered valid for unemployed job seekers.

In more detail, in the first thematic core (Exhaustion vs. Engagement) the richness and different facets of the capacity of work as self-empowerment, capable of motivating engagement in the search emerge, but so too do the risks associated with the exhaustion of energies if this search proves fruitless.

The second core (Disillusion vs. Hope) depicts the attrition and sense of helplessness, learned on the basis of the repeated failure of the search strategies adopted, that the unemployed person experiences if his or her efforts are unsuccessful. 

The third core (Cynicism vs. Trust) also points out this aspect, with a social feedback component. In this sense, both the model of Edelwich et al. and that of Maslach et al. converge in considering the influence that status and social consideration of work can have on burnout.

In the fourth core (Inefficacy vs. Efficacy in job search), respondents proposed the theme of articulating methods and strategic approach to job search. Moreover, the negative effects of their ineffectiveness appear to be exacerbated by the presence of competition among the unemployed themselves.

The results achieved with this study might be useful in providing a comprehensive key to the experiences that may unfortunately affect many people, precisely because of the effects of the COVID-19 pandemic on the world of work. In fact, although the data collected for this research predate the repeated lockdowns ordered by the Italian government (indeed, the data collection had to be interrupted because the employment centres were closed from March 2020, and it was no longer possible to interview unemployed job seekers), this four-dimensional and counter-polar model of burnout and engagement could be useful to better describe the experience of those who, when seeking employment, experience frustration and hope not only to satisfy their needs for economic subsistence and security, but also for personal self-realization through work [13,15].

## 5. Conclusions

The psychological, physical, social and economic effects of unemployment are a major public health issue, especially in light of the effects of the COVID-19 pandemic, which, globally, is continuing to have major effects on the labour market [2,65], in terms not only of unemployment, but also of job insecurity and the risk of losing significant aspects of employment [66] for large segments of the population worldwide. The results of our study can pave the way for social and public policies and interventions that allow not only for understanding and managing the inevitable and frayed discomfort resulting from job loss, but also for sustaining motivation and engagement towards finding new employment that the person can deem decent and adequate.

In fact, several methodologies and intervention programmes are now well established in the literature that can act on both sides—that of burnout and that of engagement. For example, with regard to the first aspect, there is ample evidence for the efficacy of mindfulness. This can be defined as the awareness that emerges by paying attention without judgment to the unfolding of the experience of the moment [67]. In this state, thoughts and emotions are recognized as passing events, disconnected from habitual patterns of cognitive and behavioural reactivity [68].

Mindfulness, in the different forms in which it can be implemented (e.g., individually, paired with a trainer, in groups, online [69]), facilitates clarity and insight, as well as enhancing critical reflection and effective management of challenging situations, a sense of autonomy, patience and acceptance of both pleasant and unpleasant life experiences. It reduces emotional distress and can facilitate new and more effective responses to negative behaviours and ruminating thoughts [70].

Given the psycho-physical and emotional states experienced by unemployed job seekers previously described, it is more than plausible that engaging in mindfulness activity can help them gain a realistic awareness of their contingent condition, manage the negative emotional states arising from it, formulate diverse and innovative job search strategies and manage stress in their implementation.

In a similar vein, various ways (e.g., individually, with a trainer, in groups, or online) to promote and nurture the positive polarity of work engagement have been widely documented and validated in the literature [71,72]. These include creativity learning group sessions focused on problem solving and perspective taking; active learning, role playing and social modelling [73]; virtual mindfulness sessions, homework, progress-tracking survey and e-coaching [74]; group mindfulness training, goal-setting, homework, individual e-coaching and supporting materials (e.g., web page, logbook) [75]. In this case, the perspective could be not only to manage the distress resulting from job loss, but to promote the person’s well-being and health, enabling him or her to reframe the possibilities offered by his or her contingent unemployed status.

The challenge, ultimately, is to understand if and how these same methods and intervention programmes designed for working people can be adapted and shown to be valid and useful for the unemployed as well. Their implementation could have a positive influence on the psycho-physical stress of the unemployed; of the employment services to which they turn (which currently, especially in southern Italy, are unable to adequately respond to the job demands of the unemployed); on the healthcare system (which would have fewer demands, especially in the areas of mental health and addictions); and on the conditions experienced by the families of the unemployed.

This study has some limitations. First, the sample was collected on a voluntary and convenience basis, and this may have resulted in a biased selection of interview respondents. Second, from a more strictly empirical point of view, the total corpus of text submitted for statistical analysis was quantitatively limited, although 56 interviews were conducted. This limitation, which might be corrected by increasing the number of interviews in future studies, could be at least partially justified by the respondents’ unwillingness to probe and express their experiences because of the connotations with discomfort, frustrated commitment and disillusionment. In fact, only 26.9% of the unemployed people who were approached agreed to be interviewed, and in their interviews, they often manifested an attitude almost of denunciation of their condition. Finally, this study is cross-sectional, and it will be necessary to supplement its findings with further longitudinal research, especially given the processual and evolutionary (in a degenerative sense) nature of burnout.

Despite these limitations, this line of research might be considered promising and could allow for further and more focused specification of the usefulness of the “jobless burnout” construct for unemployed job seekers, especially from the perspective of designing and implementing public health interventions focused on supporting people in this condition, aimed at psychosocial support of their “work search engagement”.

## Figures and Tables

**Figure 1 ijerph-20-05968-f001:**
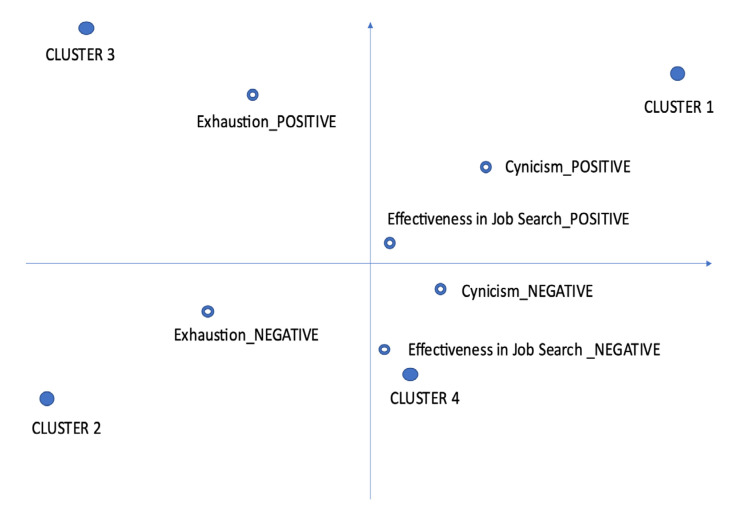
Representation of clusters and descriptive variables on the factorial plane.

**Table 1 ijerph-20-05968-t001:** Results of cluster analysis.

Cluster	% of CUs in Cluster	Label	Principal Lemmas Ordered by Decreasing Frequency English (Italian)	Frequency of Lemma Occurrence	Descriptive Variables Ordered by Decreasing Chi-Squared Value
1	21.70	Exhaustion vs. Engagement	work (lavoro)	178	Cynicism Positive: chi-squared = 259.093 Effectiveness in Job Search Positive: chi-squared= 81.933
			pleasure (piacere)	54	
			thinking (pensare)	30	
			profession (professione)	25	
2	10.83	Disillusion vs. Hope	answering (rispondere)	61	Exhaustion Negative: chi-squared = 322.29
			CV (curriculum)	41	
			sending (mandare)	24	
			announcements (annunci)	20	
			understanding (capire)	10	
			absence (assenza)	8	
			usual (solito)	8	
3	12.04	Cynicism vs. Trust	finding (trovare)	106	Exhaustion Positive: chi-squared = 294.003
			hope (speranza)	24	
			possibility (possibilità)	20	
			vigorous (energico)	12	
			positive (positivo)	11	
4	55.43	Inefficacy vs. Efficacy	searching (cercare)	76	Effectiveness in Job Search Negative: chi-squared = 88.319 Cynicism Negative: chi-squared = 8.617
			experience (esperienza)	52	
			succeeding (riuscire)	48	
			working (lavorare)	44	
			seek (ricerca)	36	
			putting (mettere)	31	
			asking (chiedere)	27	
			arriving (arrivare)	26	
			taking (prendere)	25	
			ability (capacità)	20	
			difficulty (difficoltà)	19	
			situation (situazione)	18	

## Data Availability

Data are not available due to privacy restrictions.

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
