# Peer review of "Job Seekers’ Burnout and Engagement: A Qualitative Study of Long-Term Unemployment in Italy"

_ijerph, 2023, doi:10.3390/ijerph20115968_

Round 1

Reviewer 1 Report

ijerph-2300658: The manuscript is very interesting, but some issues need further revision.

Point 1-title: The title is not precise, a precise is "Job Seekers’ Burnout and Engagement: A Qualitative Study of Long-Term Unemployment in Italy"

Point 2-intro: The introduction does not clear what general problem of job seekers’ burnout and engagement in Italy. I found the authors started with job burnout through job-seeker is invalid provide the justification of the study.

Why you did not provide general problems, gaps of theory, and objective study.

And then moved job burnout through job-seeker into the literature review section.

Point 3-method: The methods are not clear what really design is.

1. Design (what types of qualitative did you use)

2. What sample and sampling did you select the participants? For instance, why did you select 56 of 208?

3. What interview questions did you ask the participants?

4. Data analysis: Why is statistical analysis? It is conflicted in nature of qualitative methods. Variables for what? Themes or issues? Should be cleared why?

Point 4-result: The findings are too limited and invalid results.

1. What characteristic of the participants are missing in the study?

2. All quotations are invalid explanation. Should be clear what and how did you develop before starting with quotations.

From cluster 1-4.

Point 5-discussion: The study did not discuss with the main findings. Why you did not follow with each cluster and then debate with other scholars?

Point 6-conclusion: This section is too limited summarization with the main findings. Are missing practical implications, social and policy contributions.

 Point 7-refs: The texts and the list references are incorrect and un-update. Should be strictly followed the journal guidelines.

 Point 8-communication: The quality of communication is no in academic writing. May grammatical errors.

 Point 9-length: The writing styles are too long. Should be re-defined  as 7-9 lines per each paragraph is required.

Author Response

Dear Reviewer,

Thank you for your suggestions and corrections. The  attached table describes our changes to the article, based on all your comments.

Best regards,

The Authors

Reviewer 2 Report

Initially, I congratulate the authors for the way of conducting the research. The approach to the theme is current and allows for further questioning and development of other studies on the theme of “exhaustion in long-term unemployment.” The consequences of this problem are determined by how the world of work is structured (non-decent work, unemployment...), which is still highly perverse in its personal dimension and globally in different social contexts.

The textual elaboration is adequate and allows me to understand its theoretical aspects by adapting the theory of exhaustion at work - para the theme of long-term unemployment. Its empirical content was analyzed from the thematic content data of the interviews.

I offer three suggestions to the authors:

In the section – Introduction

1 - Start the introduction with subitem 1.2 Long-term unemployment. This item brings the research problem and its theoretical articulations altogether. I emphasize that it is a suggestion, and the decision corresponds to the authors' evaluation.

In the section - Materials and Methods

The qualitative analysis was conducted to express the quantitative in its qualitative nature and was adequate.

2 - I suggest the inclusion of the software specification so that comparisons of use are possible if there are no problems in disclosing the software identification.

In the section - Discussion

3- I suggest including some elements (it could be a paragraph) of the general unemployment situation for Italian workers. I emphasize that this detail is very well put in the Introduction. It could be brought here because the person's burnout is in a broader context. It's a suggestion. I hope I helped in the review process.

Final comment:

I hope to have contributed to the revision of the content of this manuscript. The present study differs from and simultaneously coherently complements the previous research by the group of authors, according to the reference presented in the manuscript. It presents essential evidence about the damage that long-term unemployment causes to human capital.

Author Response

(The authors gave the same response as above.)

Reviewer 3 Report

Thank you for the opportunity to read the manuscript. This is an interesting study, and the topic is relatively rarely discussed.

  Comments:

The title of the publication is consistent with its content. The problem and the research goal are formulated clearly and precisely. The structure of the work is correct, the research methods used do not raise objections - I am very pleased with the qualitative methodological approach to this scientific problem. The figures and tables illustrate the obtained results very well. Moreover, the authors were well aware of the limitations of this study.

  Critical note:

"Discussion" is not a discussion. Dear Authors, please try to supplement this part of the manuscript with your own interpretation and description of the significance of the results for the phenomenon under study.

Author Response

(The authors gave the same response as above.)

Reviewer 4 Report

Please, consider the following suggestions.

• Abstract is not following the guide´s indications: the results or the conclusions obtained are not clear, in addition to the number of words that the journal requires.

• Citations some are without numbers.

• Avoid redundancy pages 3, 4, 6 “and” “job” “common among”

• Data from other research are included, which can lead to confusion (Italian sample, n=208).

• On page 4, it is mentioned that there are 4 dimensions and only 3 are contemplated (the dimensions of psychophysical exhaustion and sense of ineffectiveness in the job and the dimension of cynicism)

• Aims, some suggestions about this study are proposed, the same ones that are not clear or are not very specific.

• In section 2.3, only the female sample is considered, ignoring the male one.

• The interpretation of table 1 is limited to pointing out what is mentioned in the table without there being an important contribution from the table.

• In the discussion, some data are not contrasted, such as the first core and third core: Exhaustion vs. Engagement, Cynicism vs. Trust

• The conclusions are not clear, it is limited to a review of the contributions of several authors, to the limitations and future lines of investigation of the paper.

• The information included in the conclusions fits better in the discussion

Author Response

(The authors gave the same response as above.)

Round 2

Reviewer 1 Report

Thank you for your revision. All comments are revised and suitable for publication in the IJERPH.

Author Response

We thank Reviewer#1. We have re-checked the quality of the English language of the article. We

Attach the certificate.

Author Response

Dear Reviewer #4,

Thank you for your suggestions and corrections. The table in attachment describes our changes to the article, based on all your comments.

Best regards.

The Authors
